# Anti-Browning Effect of 2-Mercaptobenzo[*d*]imidazole Analogs with Antioxidant Activity on Freshly-Cut Apple Slices and Their Highly Potent Tyrosinase Inhibitory Activity

**DOI:** 10.3390/antiox12101814

**Published:** 2023-09-29

**Authors:** Jieun Lee, Hye Soo Park, Hee Jin Jung, Yu Jung Park, Min Kyung Kang, Hye Jin Kim, Dahye Yoon, Sultan Ullah, Dongwan Kang, Yujin Park, Pusoon Chun, Hae Young Chung, Hyung Ryong Moon

**Affiliations:** 1Department of Manufacturing Pharmacy, College of Pharmacy and Research Institute for Drug Development, Pusan National University, Busan 46241, Republic of Korea; yijiun@pusan.ac.kr (J.L.); hyesoo0713@pusan.ac.kr (H.S.P.); wjd9933@pusan.ac.kr (Y.J.P.); kmk87106@pusan.ac.kr (M.K.K.); khj3358@pusan.ac.kr (H.J.K.); dahae0528@pusan.ac.kr (D.Y.); 2Department of Pharmacy, College of Pharmacy and Research Institute for Drug Development, Pusan National University, Busan 46241, Republic of Korea; hjjung2046@pusan.ac.kr (H.J.J.); hyjung@pusan.ac.kr (H.Y.C.); 3Department of Molecular Medicine, The Herbert Wertheim UF Scripps Institute for Biomedical Innovation & Technology, Jupiter, FL 33458, USA; sultanullahf@ufl.edu; 4Department of Medicinal Chemistry, New Drug Development Center, Daegu-Gyeongbuk Medical Innovation Foundation, Daegu 41061, Republic of Korea; kdw4106@kmedihub.re.kr (D.K.); pyj1016@kmedihub.re.kr (Y.P.); 5College of Pharmacy and Inje Institute of Pharmaceutical Sciences and Research, Inje University, Gimhae 50834, Republic of Korea; pusoon@inje.ac.kr

**Keywords:** browning, melanin, tyrosinase, antioxidant, 2-mercaptobenzo[*d*]imidazole, docking simulation

## Abstract

Ten 2-mercaptobenzimidazole (2-MBI) analogs were synthesized as potential tyrosinase inhibitors because mercapto-containing compounds can bind to copper ions at the active site of tyrosinase to inhibit enzyme activity. Nine 2-MBI analogs showed sub-micromolar IC_50_ values for mushroom tyrosinase monophenolase activity; analog **4** was 280-fold more potent than kojic acid, and in diphenolase activity, **6** was 970-fold more potent than kojic acid. The inhibition mode of the 2-MBI analogs was investigated using kinetic studies supported by docking simulations. Benzimidazoles without the 2-mercapto substituent of the 2-MBI analogs lost their tyrosinase inhibitory activity, implying that the 2-mercapto substituent plays an important role in tyrosinase inhibition. The 2-MBI analogs exerted potent antioxidant effects against 2,2′-azino-bis(3-ethylbenzothiazoline-6-sulfonic acid (ABTS), 2,2-diphenyl-1-picrylhydrazyl (DPPH), and reactive oxygen species (ROS). The results obtained from apple slices and human embryonic kidney cells (HEK-293) suggest that most 2-MBI analogs are sufficiently safe candidates to delay the browning of apple slices effectively. Thus, these results support the potential use of 2-MBI analogs as anti-browning agents in foods such as mushrooms, vegetables, and fruits.

## 1. Introduction

Melanins are a group of polymeric pigments with complex chemical structures widely distributed in most living organisms, including various species ranging from bacteria to humans [1]. In humans, melanin is the primary factor that determines skin color [2] and is synthesized in melanosomes, melanin-containing lysosomal-related organelles of melanocytes present at the bottom of the innermost layer of the epidermis [3]. In addition to protecting cells from UV radiation [4], melanin can scavenge hazardous chemicals and free radicals, including reactive oxygen species (ROS) [5], and contribute to defense mechanisms (innate immune response) in insects [6] and brain neuron protection [7]. It also strengthens virulence mechanisms in organisms such as bacteria and fungi [8].

Consumers value quality when purchasing crops, such as vegetables and fruits, and the quality of crops is closely related to their browning. Tyrosinase is a glycosylated metalloenzyme containing two copper ions that play an important role in crop browning. It acts as a multifunctional enzyme, a monophenolase, and a diphenolase, which hydroxylates l-tyrosine to afford l-dopa, which oxidizes l-dopa to produce dopaquinone, which is, in turn, converted into melanin, the final product, by various chemico–enzymatic reactions [9,10]. This process, called browning, browns crops, develops undesirable flavors, and causes nutrient loss, lowering crop quality. Thus, a delay in or inhibition of crop browning can increase the value and quality of crops [11]. Because of its role as a rate-determining enzyme in melanin production, tyrosinase has attracted attention in the food industry as an important means of preventing the browning of crops.

Melanin is divided into eumelanin and pheomelanin, which are biosynthesized from a common substrate, l-tyrosine, via enzymatic and non-enzymatic reactions. l-Tyrosine is converted into dopaquinone by the successive monophenolase and diphenolase activities of tyrosinase. The fate of dopaquinones depends on the presence or absence of mercapto-containing (thiol-containing) compounds (e.g., glutathione and cysteine). In the presence of mercapto-containing compounds, dopaquinone is converted to pheomelanin, responsible for the red/yellow color. In contrast, in the absence of mercapto-containing compounds, it is converted to eumelanin, which is responsible for the black/brown color. Therefore, mercapto-containing compounds can influence the biosynthetic ratio of eumelanin to pheomelanin [12]. In addition, mercapto-containing compounds have been reported to bind copper ions at the active site of tyrosinase and inhibit its enzymatic activity [13]. On the other hand, it has been suggested that the antioxidant ability of a compound may inhibit melanin production by suppressing oxidation processes (l-tyrosine to l-dopa and l-dopa to dopaquinone) during melanogenesis [14,15]. Mercapto-containing compounds can also act as antioxidants that scavenge free radicals, including ROS [16]. Thus, because of their potential to act as (i) modifiers of the eumelanin/pheomelanin ratio, (ii) tyrosinase inhibitors through copper ion chelation, and/or (iii) radical scavengers or antioxidants, mercapto-containing compounds have been considered potential melanin-inhibitory agents to prevent or retard browning in crops.

In a previous study of more than 15 years, novel tyrosinase inhibitors were explored using a PUSC (β-phenyl-α,β-unsaturated carbonyl) scaffold, and demonstrated that this scaffold plays a key role in conferring tyrosinase activity [17,18,19,20]. Compounds with the PUSC scaffold showed potent tyrosinase and anti-melanogenic effects in mushroom and mammalian tyrosinases. However, they could not be applied to anti-browning experiments on crops due to their low water solubility. In contrast, Lee et al. reported that among eight phenolic acids, kojic acid, a well-known tyrosinase inhibitor, showed the strongest inhibitory effect on the browning of apple slices [21]. Peng et al. reported that kojic acid and kojic acid-1,3,4-oxadiazoles exhibited anti-browning effects on freshly cut mushrooms [22]. Based on these findings, water-soluble 2-mercaptobenzo[*d*]imidazole (2-MBI) compounds were synthesized to identify more potent and water-soluble anti-browning agents than kojic acid. Typically, fruits undergo a slow process of oxidation after harvest, which produces undesirable flavors and often reduces the quality of the fruit. Due to the antioxidant ability of the mercapto functional group, applying 2-MBI analogs to fruits is likely to not only suppress browning by inhibiting tyrosinase activity, but also help improve fruit quality through antioxidant effects.

## 2. Materials and Methods

### 2.1. Chemistry

#### 2.1.1. General Methods

All reagents, chemicals, and solvents were purchased commercially, and known dry methods (distillation over CaH_2_ or Na/benzophenone) were used for anhydrous solvents. The progress of the reaction was determined by monitoring thin-layer chromatography (TLC) (Silica gel 60 F_254_). The reaction mixture was purified using flash column chromatography using MP Silica (60 Å). ^1^H- and ^13^C-NMR (nuclear magnetic resonance) spectra were measured using a Varian Unity AS500 unit (Agilent Technologies, Santa Clara, CA, USA), a JEOL ECZ400S spectrometer (Tokyo, Japan), and a Varian Unity INOVA 400 instrument (Agilent Technologies, Santa Clara, CA, USA). Coupling constants (*J*) and chemical shifts (*δ*) were recorded in hertz (Hz) and in parts per million (ppm), respectively. The splitting patterns are presented as broad singlet (brs), broad multiplet (brm), singlet (s), doublet (d), doublet of doublets (dd), doublet of doublets of doublets (ddd), triplet (t), and multiplet (m). ^1^H and ^13^C NMR spectroscopy data of 2-MBI analogs **1**–**10** are presented in the Appendix A.

#### 2.1.2. Synthesis of 2-MBI Analogs **1**–**10**

Two synthetic methods were used for the synthesis of analogs **1**–**10**.

##### General Procedure for the Synthesis of **1**, **2**, **6**, **7**, and **9**

To a stirred solution of *o*-phenylenediamines (1,2-phenylenediamine for **1**, 3,4-diaminotoluene for **2**, 3,4-diaminobenzophenone for **6**, 4-nitro-1,2-phenylenediamine for **7**, or 2,3-diaminotoluene for **9**) and NaOH (1.16 equiv.) in ethanol (1.1 mL/L mmol of *o*-phenylenediamines) and H_2_O (0.16 mL/1 mmol of *o*-phenylenediamines) was added CS_2_ (1.16 equiv.). After the reaction mixture was heated at 80 °C for 1–9 h, charcoal (1.5 g/L mmol of *o*-phenylenediamines) was added and heated at 80 °C for 10 min. The hot reaction mixture was filtered and washed with hot water (60–70 °C), and the pH of filtrate was adjusted to 2 using 50% acetic acid. The reaction mixture was stored in the refrigerator overnight and the generated solid was filtered and washed with water to afford pure 2-mercaptobenzimidazoles (**1**, 86%; **2**, 83%; **6**, 77%; **7**, 76%; **9**, 68%) as solids.

##### General Procedure for the Synthesis of **3**–**5** and **8**–**10**

A solution of *o*-phenylenediamines (4,5-dimethyl-1,2-phenylenediamine for **3**, 4-chloro-1,2-phenylenediamine for **4**, 4-methoxy-1,2-phenylenediamine for **5**, 4-fluoro-1,2-phenylenediamine for **8**, 2,3-diaminotoluene for **9**, or 3,4-diaminobenzonitrile for **10**) and sodium *N*,*N*-diethyldithiocarbamate trihydrate (2.5 equiv.) in DMF (4 mL/L mmol of *o*-phenylenediamines) was heated in the presence of AlCl_3_ (0.1 equiv.) at 120 °C for 0.5–9 h. The work-up process to obtain the desired product is as follows. For **3**, after adding water, the generated solid was filtered to afford **3** (75%). For **4**, after volatiles were evaporated under reduced pressure and water was added, the resultant solid was filtered and washed with hexane and dichloromethane (2:1) to give **4** (88%). For **5**, **8**, and **9**, after partitioning between ethyl acetate and H_2_O, the organic layer was dried and evaporated. The solid residue was filtered and washed with hexane and dichloromethane (2:1) and ethyl acetate to give **5** (92%) as a solid or the solid residue was filtered and washed with dichloromethane to give **8** (52%) and **9** (32%) as solids. For **6**, after volatiles were evaporated, the resulting residue was filtered and washed with hexane and dichloromethane (1:1). The filter cake was purified by silica gel column chromatography using hexane and ethyl acetate (2:1) as eluent to give **6** (83%) as a solid. For **10**, volatiles were evaporated under reduced pressure and the resulting residue was partitioned between ethyl acetate and water. The organic layer was evaporated and the residue was filtered and washed using hexane and dichloromethane (1:1) to afford **10** (85%).

#### 2.1.3. NMR Data for 2-MBI Analogs **1**–**10**

^1^H and ^13^C NMR spectra (Appendix A) of 2-MBI analogs **1**–**10** are presented in the Appendix A.

1*H*-Benzo[*d*]imidazole-2-thiol (**1**)

^1^H NMR (400 MHz, DMSO-*d*_6_) *δ* 12.47 (s, 1H, NH), 7.12–7.04 (m, 4H, 4-H, 5-H, 6-H, 7-H); ^13^C NMR (100 MHz, DMSO-*d*_6_) *δ* 168.7, 132.8, 122.8, 110.0.

5-Methyl-1*H*-benzo[*d*]imidazole-2-thiol (**2**)

^1^H NMR (400 MHz, DMSO-*d*_6_) *δ* 12.35 (s, 1H, NH), 6.97 (d, 1H, *J* = 8.0 Hz, 7-H), 6.90 (s, 1H, 4-H), 6.88 (d, 1H, *J* = 8.0 Hz, 6-H), 2.29 (s, 3H, CH_3_); ^13^C NMR (100 MHz, DMSO-*d*_6_) *δ* 168.3, 133.0, 132.2, 130.7, 123.7, 110.1, 109.6, 21.5.

5,6-Dimethyl-1*H*-benzo[*d*]imidazole-2-thiol (**3**)

^1^H NMR (400 MHz, DMSO-*d*_6_) *δ* 12.25 (brs, 1H, NH), 6.88 (s, 2H, 4-H, 7-H), 2.18 (s, 6H, 2 × CH_3_); ^13^C NMR (100 MHz, DMSO-*d*_6_) *δ* 167.7, 131.1, 131.0, 110.6, 20.1.

5-Chloro-1*H*-benzo[*d*]imidazole-2-thiol (**4**)

^1^H NMR (400 MHz, DMSO-*d*_6_) *δ* 7.18–7.10 (m, 3H, 4-H, 6-H, 7-H); ^13^C NMR (100 MHz, DMSO-*d*_6_) *δ* 174.6, 138.5, 136.5, 132.0, 127.5, 115.8, 114.5. 

5-Methoxy-1*H*-benzo[*d*]imidazole-2-thiol (**5**)

^1^H NMR (400 MHz, DMSO-*d*_6_) *δ* 12.38 (brs, 1H, NH), 7.03 (d, 1H, *J* = 8.4 Hz, 7-H), 6.72 (dd, 1H, *J* = 8.4, 2.4 Hz, 6-H), 6.67 (d, 1H, *J* = 2.4 Hz, 4-H) 3.74 (s, 3H, OCH_3_); ^13^C NMR (100 MHz, DMSO-*d*_6_) *δ* 168.3, 156.3, 133.6, 126.9, 110.5, 110.3, 95.0, 56.1.

(2-Mercapto-1*H*-benzo[*d*]imidazol-5-yl)(phenyl)methanone (**6**)

^1^H NMR (400 MHz, DMSO-*d*_6_) *δ* 7.65 (d, 2H, *J* = 7.6 Hz, 2′-H, 6′-H), 7.61 (t, 1H, *J* = 7.6 Hz, 4′-H), 7.54–7.47 (m, 3H, 7-H, 3′-H, 5′-H), 7.44 (s, 1H, 4-H), 7.23 (d, 1H, *J* = 8.4 Hz, 6-H); ^13^C NMR (100 MHz, DMSO-*d*_6_) *δ* 195.6, 171.2, 138.4, 137.5, 133.4, 132.6, 131.1, 129.9, 129.0, 125.6, 111.4, 109.8.

5-Nitro-1*H*-benzo[d]imidazole-2-thiol (**7**)

^1^H NMR (400 MHz, DMSO-*d*_6_) *δ* 8.03 (d, 1H, *J* = 8.8 Hz, 6-H), 7.85 (s, 1H, 4-H), 7.25 (d, 1H, *J* = 8.8 Hz, 7-H); ^13^C NMR (100 MHz, DMSO-*d*_6_) *δ* 172.3, 143.1, 137.9, 132.8, 119.5, 109.8, 105.2. 

5-Fluoro-1*H*-benzo[*d*]imidazole-2-thiol (**8**)

^1^H NMR (400 MHz, DMSO-*d*_6_) *δ* 7.07 (ddd, 1H, *J* = 8.4, 4.8, 0.8 Hz), 6.95– 6.88 (m, 2H); ^13^C NMR (100 MHz, DMSO-*d*_6_) *δ* 169.2, 159.1 (d, *J* = 235.8 Hz), 133.1 (d, *J* = 13.2 Hz), 129.2, 110.9 (d, *J* = 9.8 Hz), 110.1 (d, *J* = 24.7 Hz), 97.6 (d, *J* = 28.0 Hz).

4-Methyl-1*H*-benzo[*d*]imidazole-2-thiol (**9**)

^1^H NMR (400 MHz, DMSO-*d*_6_) *δ* 6.99 (t, 1H, *J* = 7.6 Hz, 6-H), 6.94 (d, 1H, *J* = 7.6 Hz, 7-H), 6.89 (d, 1H, *J* = 7.6 Hz, 5-H), 2.36 (s, 3H, CH_3_); ^13^C NMR (100 MHz, DMSO-*d*_6_) *δ* 167.9, 132.3, 131.9, 124.0, 123.1, 120.4, 107.7, 16.7. 

2-Mercapto-1*H*-benzo[*d*]imidazole-5-carbonitrile (**10**)

^1^H NMR (400 MHz, DMSO-*d*_6_) *δ* 7.52–7.48 (m, 2H, 4-H, 6-H), 7.22 (d, 1H, *J* = 8.8 Hz, 7-H); ^13^C NMR (100 MHz, DMSO-*d*_6_) *δ* 171.2, 136.2, 132.9, 127.5, 119.8, 113.4, 110.7, 104.8.

### 2.2. Mushroom Tyrosinase Inhibitory Assay

The inhibitory activities of analogs **1**–**10** against mushroom tyrosinase were evaluated according to a previously described protocol [23] with minor modifications. The effect of the analogs on enzyme inhibition was investigated in the presence of l-dopa and l-tyrosine. Briefly, an aliquot (20 µL, 1000 units/mL) of mushroom tyrosinase (aqueous solution) was added to each well of a 96-well microplate containing 170 µL of substrate mixture (345 μM l-dopa or l-tyrosine plus 17.2 mM phosphate buffer [pH 6.5]) and 10 µL of **1**–**10** with 3–5 different concentrations or 10 µL of kojic acid (positive reference control). After incubating the assay mixture for 0.5 h at 37 °C, the amounts of dopachrome produced during the incubation time were evaluated by measuring the absorbances of each well at 475 nm using a VersaMax^TM^ microplate reader (Molecular Devices, Sunnyvale, CA, USA).

### 2.3. Kinetic Studies of Tyrosinase Inhibition

For kinetic studies of mushroom tyrosinase inhibition by the 2-MBI analogs (**4**, **6**, and **10**) showing excellent tyrosinase inhibitory activity, Lineweaver–Burk plots were obtained in the presence of l-dopa as the substrate. Briefly, a dimethyl sulfoxide (DMSO) solution (10 µL) containing test analogs (final concentrations: 0, 5, 10, and 20 nM) was added to the wells of a 96-well plate containing mushroom tyrosinase (20 µL, 200 units/mL) and an aqueous solution (170 µL) consisting of l-dopa (final concentrations: 0.125, 0.25, 0.5, 1, 2, 4, and 8 mM) and phosphate buffer (final concentration: 14.7 mM, pH 6.5). The increase in the absorbance at 475 nm (ΔOD_475_/min) was measured using a VersaMax^TM^ microplate reader to determine the initial rate of dopachrome generation. The maximal velocity (V_max_) was determined from the y-axis intercept values of the Lineweaver–Burk plots obtained at seven different l-dopa concentrations. Dixon plots were obtained from Lineweaver–Burk plots for inhibitors by plotting the reciprocal of the reaction velocity against the inhibitor concentration.

### 2.4. In Silico Docking Simulation

#### 2.4.1. Docking Simulation Using Schrodinger Suite

In silico docking simulations were conducted to examine the chemical interactions between the 2-MBI analogs and mushroom tyrosinase. The simulations were conducted using two docking simulation programs, Schrödinger Suite (2021-2) and AutoDock Vina 1.1.2 (software developed by The Scripps Research Institute), according to previous protocols [24], with minor modifications. The X-ray crystal structure of mushroom tyrosinase (*A. bisporus*, Protein Data Bank (PDB) ID: 2Y9X) was obtained from PDB and was imported from PDB into Maestro 12.4′s Protein Preparation Wizard. Unnecessary protein chains were deleted from the mushroom tyrosinase crystal structure. The crystal structure was optimized by adding hydrogen atoms and removing >3 Å water molecules from the enzyme. The binding location of tropolone, the ligand of the imported X-ray crystal structure, was used to determine the glide grid of the mushroom tyrosinase active site [25,26]. The structures of in silico docking compounds (**1**, **2**, and kojic acid, the positive reference compound) were imported into Maestro’s entry list in CDXML format and developed using LigPrep prior to ligand docking. Using the Glide task list, chemical structures were docked to the tyrosinase glide grid [27]. Binding affinities and protein–ligand interactions were determined using the glide standard precision method.

#### 2.4.2. Docking Simulation Using AutoDock Vina

According to a previous protocol [28], docking simulations of analogs **1**–**10** were performed using AutoDock Vina 1.1.2. The 3D structures of **1**–**10** were prepared using Chem3D Pro 12.0 software, and kojic acid was used as a positive reference material. The three-dimensional X-crystal structure of tyrosinase (*A. Bisporus*) was obtained from the PDB (ID: 2Y9X). The binding scores between the docking compounds (analogs **1**–**10** and kojic acid) and tyrosinase were calculated using AutoDock Vina and Chimera 35.91.958. LigandScout 4.4 was used to create pharmacophore figures showing possible interactions between the amino acids of tyrosinase and the ligands (analogs **1**–**10** and kojic acid).

### 2.5. 2,2-Diphenyl-1-picrylhydrazyl (DPPH) Radical Scavenging Assay

The DPPH radical scavenging activities of analogs **1**–**10** were evaluated according to a previously described protocol [29] to examine the antioxidant effects of analogs **1**–**10**. Briefly, an aliquot (180 µL) of DPPH methanol solution (0.2 mM) was mixed with a 2-MBI analogs’ DMSO (dimethyl sulfoxide) solution in each well of a 96-well plate, and the mixture was left in the dark for 0.5 h at room temperature. The optical density of each well was analyzed using a microplate reader (VersaMax^®^, Molecular Devices, Sunnyvale, CA, USA).

### 2.6. 2,2′-Azino-bis(3-ethylbenzothiazoline-6-sulfonic Acid (ABTS) Radical Cation Scavenging Assay

The ABTS radical cation-scavenging activities of analogs **1**–**10** were investigated according to a previously described protocol [30] with minor modifications. Briefly, to prepare a test solution for the ABTS radical cation scavenging activity, potassium persulfate aqueous solution (10 mL, 2.45 mM) was mixed with ABTS aqueous solution (10 mL, 7 mM) and left for 12 h at room temperature in the dark until the absorbance remained constant. Prior to the ABTS radical cation scavenging assay experiments, the absorbance at 734 nm was adjusted to 0.70 ± 0.02 by diluting the ABTS radical cation solution with water. An aliquot (10 µL) of **1**–**10** or Trolox (the positive reference material) at 100 µM was mixed with the ABTS radical cation solution (90 µL) in each well of a 96-well plate and incubated for 10 min at ambient temperature in the dark. The optical density of each well was measured at 734 nm using a VersaMax^®^ microplate reader. All experiments were performed independently 3 times.

### 2.7. In Vitro Reactive Oxygen Species (ROS) Scavenging Assay

The scavenging effects of analogs **1**–**10** on in vitro ROS were evaluated using 2′,7′-dichlorodihydrofluorescein diacetate (DCFH-DA), the oxidant-sensitive fluorescence probe [31]. An aliquot (10 µL) of **1**–**10** (final concentration: 40 µM) or Trolox (final concentration: 40 µM, the positive reference control) was added to each well in a black 96-well plate. Then, 10 µL of SIN-1 (10 µM, 3-morpholinosydnonimine), 180 µL of PBS (phosphate-buffered saline, pH 7.4) solution, and a mixed solution (50 µL) containing 600 units of esterase and 12.5 µM DCFH-DA were successively added to each well. At emission and excitation wavelengths of 535 and 485 nm, fluorescence intensity was measured using a fluorescence microplate reader (Berthold Technologies GmbH & Co., Wien, Austria).

### 2.8. Anti-Browning Assay of Freshly-Cut Apple Slices

#### 2.8.1. Sample Preparation

Apples (*Malus pumila* Miller) of similar maturity were purchased from a local market (Emart24^®^, Busan, Republic of Korea). The apples were cut into small equal-sized pieces (W × D × H, 1.5 ± 0.1 cm × 1.5 ± 0.1 cm × 0.3 ± 0.05 cm). Three apple slices were placed in glass dishes with a diameter of 5 cm.

#### 2.8.2. Sample Treatment

Each apple slice was sprayed with control ultrapure water or a 5 mM test sample solution (kojic acid, phenylthiourea (PTU), or sodium salts of analogs **1**, **2**, and **4**–**10**) at 0, 12, 24, and 36 h, and browning results were observed for 48 h. The apple slice dishes were placed in an incubator maintained at 20 °C. Kojic acid and PTU were used as positive controls. All samples, except PTU, were prepared as aqueous solutions. PTU test samples were prepared in a DMSO solution as it is only soluble in water at very low concentrations.

#### 2.8.3. Browning Color Measurement

At five time points (0, 12, 24, 36, and 48 h), apple slices were photographed using a camera (iPhone 11), and browning colors (a*, b*, and L*) were measured using a CR-10 spectrophotometer (Konica Minolta Sensing, Inc., Osaka, Japan). In the spectrophotometer, the colors were described by a* (redness and greenness), b* (yellowness and blueness), and L* (lightness) values according to the Commission Internationale de l’Eclairage (CIE) color system. The overall color difference (Δ*E*) was calculated using the following formula: (Δ*E*)^2^ = (a − a_initial_)^2^ + (b − b_initial_)^2^ + (L − L_initial_)^2^. In addition, based on the photographs, the relative browning intensities were calculated using CS analyzer 3.2 image analysis software.

### 2.9. Human Embryonic Kidney Cells (HEK-293) Cell Culture

HEK-293 cells were obtained from American Type Culture Collection (ATCC; Manassas, VA, USA). Trypsin, streptomycin, fetal bovine serum (FBS), phosphate-buffered saline (PBS), penicillin, and Dulbecco’s modified Eagle’s medium (DMEM) for HEK-293 cell culture were purchased from Thermo Fisher Scientific (Carlsbad, CA, USA). HEK-293 cells were cultured in a DMEM solution containing penicillin (100 IU/mL), 10% heat-inactivated FBS, and streptomycin (100 µg/mL) in a 5% CO_2_ environment at 37 °C.

### 2.10. Cell Viability Assays

Cytotoxicity assays were performed using an EZ-Cytox assay (EZ-1000, DoGenBio, Seoul, Republic of Korea) in HEK-293 cells [22]. Briefly, 96-well plates containing HEK-293 cells at a density of 1 × 10^4^ cells per well were cultured in a 5% CO_2_ environment at 37 °C for 24 h. After 20 h, cells were treated with 2-MBI analogs **1**−**10** at 4 different concentrations (final concentration: 0, 2.5, 10, and 40 µM) and incubated for 24 h at 37 °C in a 5% CO_2_ environment. Then, 10 µL aliquot of the EZ-Cytox solution was added to each well. After 2 h incubation at 37 °C, the absorbance of each well was measured using a microplate reader (VersaMax^®^; Molecular Devices) at 450 nm.

### 2.11. Statistical Analysis

The experiments were performed in triplicates. Data are expressed as mean (±standard errors of the mean (SEM)), and one-way analysis of variance (ANOVA) followed by a Bonferroni post hoc test determined significant differences between the groups. GraphPad Prism 5 software (La Jolla, CA, USA) was used for all analyses. Two-sided *p*-values < 0.05 were considered statistically significant.

## 3. Results and Discussion

### 3.1. Synthesis of Analogs ***1***–***10***

Ten 2-MBI analogs (**1**–**10**) were synthesized using the two synthetic methods shown in Figure 1. *o*-Phenylenediamine was reacted with carbon disulfide in the presence of sodium hydroxide at reflux or with sodium *N*,*N*-diethylthiocarbamate in the presence of AlCl_3_ at reflux in *N*,*N*-dimethylformamide (DMF) to produce analogs **1**–**10**. The structures of **1**–**10** were easily analyzed and confirmed using ^1^H and ^13^C NMR spectroscopy (Appendix A).

### 3.2. Inhibitory Activities of Analogs ***1***–***10*** against Mushroom Tyrosinase

First, the inhibitory effects of **1**–**10** on the diphenolase activity of mushroom tyrosinase were evaluated in vitro using l-dopa as a substrate. Kojic acid was used as a positive control. All 2-MBI analogs strongly inhibited diphenolase activity in a concentration-dependent manner. The IC_50_ values are listed in Table 1. All analogs inhibited diphenolase activity much stronger than kojic acid (IC_50_ = 19.52 ± 0.68 μM), and these analogs exhibited 26–970-fold more potent inhibitory activities than kojic acid. All analogs showed sub-micromolar IC_50_ values; **6** (IC_50_ = 0.02 ± 0.01 μM) with a 5-benzoyl substituent was the strongest diphenolase inhibitor. Analog **1**, with no substituents, also showed potent inhibitory activity (IC_50_ = 0.15 ± 0.00 μM). The insertion of an electron-donating group (EDG) produced different results depending on the insertion position. Insertion of EDG at position 5 of the 2-MBI ring slightly enhanced the diphenolase inhibitory activity. In contrast, the insertion of EDG at position 4 decreased the inhibitory activity: Analog **2** (IC_50_ = 0.10 ± 0.00 μM) with a 5-methyl substituent and analog **5** (IC_50_ = 0.08 ± 0.03 μM) with a 5-methoxyl substituent had slightly lower IC_50_ values than analog **1**. However, analog **9** with a 4-methyl substituent had a higher IC_50_ value (IC_50_ = 0.74 ± 0.17 μM) than analog **1**. On the other hand, the additional methyl substituent at position 6 of analog **2** reduced the diphenolase inhibitory activity (IC_50_ of analog **3**: 0.69 ± 0.04 μM). Introduction of the electron-withdrawing group (EWG) at position 5 of analog **1** inhibited diphenolase activity more strongly than **1**: Analog **4** (IC_50_ = 0.03 ± 0.00 μM) with a 5-chloro substituent, analog **6** (IC_50_ = 0.02 ± 0.01 μM) with a benzoyl substituent, analog **7** (IC_50_ = 0.09 ± 0.01 μM) with a 5-nitro substituent, analog **8** (IC_50_ = 0.07 ± 0.01 μM) with a 5-fluoro substituent, and analog **10** (IC_50_ = 0.04 ± 0.02 μM) with a 5-cyano substituent.

Second, the inhibitory effects of **1**–**10** on mushroom tyrosinase monophenolase were evaluated using l-tyrosine as a substrate. As in the diphenolase inhibitory test, kojic acid was used as the positive reference control. All analogs exhibited strong monophenolase inhibitory activity in a concentration-dependent manner. All analogs (**1**–**10**) inhibited monophenolase activity slightly less than diphenolase activity (Table 1). However, all analogs inhibited monophenolase activity much stronger than kojic acid (IC_50_ = 16.83 ± 3.46 μM), and most analogs exhibited 20–280-fold more potent inhibitory activities than kojic acid. All analogs had sub-micromolar IC_50_ values except for analog **9**. Analog **4** (IC_50_ = 0.06 ± 0.01 μM), with a 5-chloro substituent, was the most potent monophenolase inhibitor. Analog **1,** with no substituent on the 2-MBI ring, inhibited monophenolase activity with an IC_50_ value of 0.45 ± 0.10 μM. The effects of the substituents introduced on the 2-MBI ring on the monophenolase activity were very similar to those observed for diphenolase. Insertion of the EDG and EWG at position 5 of the 2-MBI ring enhanced the monophenolase inhibitory activity. Generally, the EWG showed slightly higher inhibitory activity than the EDG: **2** (IC_50_ = 0.16 ± 0.02 μM) with a 5-methyl substituent and **5** (IC_50_ = 0.18 ± 0.01 μM) with a 5-methoxyl substituent vs. **4** (IC_50_ = 0.06 ± 0.01 μM) with a 5-chloro substituent, **6** (IC_50_ = 0.10 ± 0.01 μM) with a benzoyl substituent, **7** (IC_50_ = 0.14 ± 0.01 μM) with a 5-nitro substituent, **8** (IC_50_ = 0.19 ± 0.02 μM) with a 5-fluoro substituent, and **10** (IC_50_ = 0.11 ± 0.01 μM) with a 3-cyano substituent. On the other hand, the insertion of a methyl substituent at position 4 of the 2-MBI ring in analog **1** diminished the monophenolase inhibitory activity by more than 10-fold (IC_50_ of analog **9** = 5.78 ± 0.43 μM). As observed in diphenolase inhibition experiments, the additional 6-methyl substituent in analog **2** reduced the monophenolase inhibitory activity four times (IC_50_ of analog **3** = 0.76 ± 0.03 μM). These results indicated that 2-MBI analogs are extremely potent tyrosinase inhibitors that can prevent the browning of crops in the food industry.

**Table 1 antioxidants-12-01814-t001:** The IC_50_ values of 2-mercaptobenzimidazole (2-MBI) analogs **1**–**10** against monophenolase and diphenolase of mushroom tyrosinase.

Compd	Structure	IC_50_ (μM)		Compd	Structure	IC_50_ (μM)	
		Monophenolase	Diphenolase			Monophenolase	Diphenolase
**1**	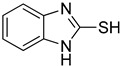	0.45 ± 0.10	0.15 ± 0.00	**7**	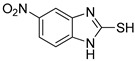	0.14 ± 0.01	0.09 ± 0.01
**2**	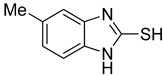	0.16 ± 0.02	0.10 ± 0.00	**8**	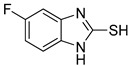	0.19 ± 0.02	0.07 ± 0.01
**3**	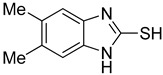	0.76 ± 0.03	0.69 ± 0.04	**9**	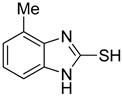	5.78 ± 0.43	0.74 ± 0.17
**4**	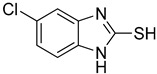	0.06 ± 0.01	0.03 ± 0.00	**10**	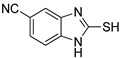	0.11 ± 0.01	0.04 ± 0.02
**5**	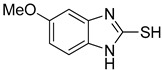	0.18 ± 0.01	0.08 ± 0.03	*^a^*KA	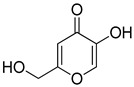	16.83 ± 3.46	19.52 ± 0.68
**6**	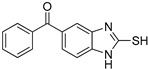	0.10 ± 0.01	0.02 ± 0.01				

*^a^*KA means kojic acid.

### 3.3. Mode of Action of 2-Mercaptobenzimidazole (2-MBI) Analogs

Kinetic studies of mushroom tyrosinase inhibition were performed using **4**, **6**, and **10**, with IC_50_ values lower than 50 nM, using l-dopa as a substrate to investigate the mechanism of tyrosinase inhibition by the 2-MBI analogs. Kinetic analyses were performed at seven different concentrations (0.125–8 mM) of l-dopa and four concentrations (0, 5, 10, and 20 nM) of the inhibitors (**4**, **6**, and **10**), resulting in Lineweaver–Burk plots for each inhibitor. Each Lineweaver–Burk plot for inhibitors **6** and **10** created four straight lines merged at a point on the y-axis (Figure 1), showing that V_max_ was consistent regardless of the inhibitor concentration. K_M_ increased with increasing inhibitor concentration. In contrast, the Lineweaver–Burk plot for inhibitor **4** revealed that four straight lines merged at a point in the second quadrant, demonstrating that V_max_ decreased and K_M_ increased with increasing inhibitor concentrations. These results indicated that analogs **6** and **10** were competitive mushroom tyrosinase inhibitors, whereas analog **4** was a mixed-type inhibitor.

Dixon plots are plotted as the reciprocal of the reaction velocity versus the inhibitor concentration. Each Lineweaver–Burk plot for the inhibitors was converted into corresponding Dixon plots (Figure 2) to determine the inhibition constant (K_i_) for the inhibitor–enzyme complex. Each Dixon plot provided four straight lines merged at a point in the second quadrant, and each vertical line from each convergence point to the x-axis provided the −K_i_ values of the inhibitors. The K_i_ values for the inhibitor–mushroom tyrosinase complex were 9.4, 8.6, and 14.8 nM for **4**, **6**, and **10**, respectively, indicating that **6** formed the strongest inhibitor–tyrosinase complex, followed by **4** and **10**.

### 3.4. In Silico Docking Simulation of 2-Mercaptobenzimidazole (2-MBI) Analogs Using Mushroom Tyrosinase

A docking simulation was performed using mushroom tyrosinase to examine the interactions between 2-MBI analogs (**1** and **2**) and amino acid residues at the active site of tyrosinase. The X-ray crystal structure of mushroom tyrosinase was obtained from the Protein Data Bank (PDB). The crystal structure of *Agaricus bisporus* tyrosinase with PDB ID 2Y9X was utilized for the docking simulation of 2-MBI analogs using the Schrodinger Suite. Kojic acid was used as the positive control. Figure 3 shows the binding interactions between the docking compounds and tyrosinase in three-dimensional (3D) and two-dimensional (2D) formats.

All the ligands (kojic acid, **1**, and **2**) bound to the tyrosinase active site. Kojic acid interacted with tyrosinase via three interactions: pi–pi stacking between the 4-pyranone and His263, a hydrogen bond between the 5-hydroxyl and Met280, and metal coordination between a 2-hydroxymethyl and Cu401. These interactions provided a binding affinity of −4.187 kcal/mol. For analog **1**, the imidazole ring of benzimidazole interacted with two amino acids, His85 and His259, via pi–pi stacking interactions, resulting in a binding affinity of −5.394 kcal/mol. As observed in analog **1**, **2** also interacted with the same amino acid residues of tyrosinase via pi–pi stacking, affording the binding affinity of −5.615 kcal/mol. Contrary to the expectations of this study, the 2-mercapto substituent of the analogs did not contribute to the ligand–enzyme interactions. Therefore, a docking simulation was performed using AutoDock Vina. The docking results are shown in Figure 4 for kojic acid, **1**, **2**, and **6** and Appendix A for **3**–**5** and **7**–**10**.

In the in silico docking simulation using AutoDock Vina, kojic acid interacted with tyrosinase via three interactions: pi–pi stacking between the 4-pyranone and His263 and two hydrogen bonds between the 5-hydroxyl and Met280 and between the 2-hydroxymethyl and three amino acids (His61, His263, and His296), providing the binding affinity of −5.4 kcal/mol (Figure 4). For analog **1**, the benzene ring of the 2-MBI ring interacted with two amino acids, Val283 and Ala286, via hydrophobic interactions, giving a binding affinity of −5.5 kcal/mol, similar to that of kojic acid. Analog **2** generated three hydrophobic interactions: the benzene ring of the 2-MBI ring in **2** interacted with Val283 and Ala286, similar to analog **1**, and the methyl substituent formed a hydrophobic interaction with Phe292. These interactions provided a binding affinity of −5.3 kcal/mol, similar to kojic acid. In **6**, the benzene ring of the 2-MBI ring and the phenyl ring of the benzoyl group participated in hydrophobic interactions. The benzene ring of the 2-MBI ring interacted with Val248 and Phe264, and the phenyl ring of the benzoyl group interacted with Val283 and Ala286. In addition, the phenyl ring in **6** formed pi–pi stacking with His263. These interactions provided analog **6** with the strongest binding affinity of −7.5 kcal/mol, consistent with the inhibition of mushroom tyrosinase activity.

As shown in Appendix A, the remaining analogs, **3**–**5** and **7**–**10**, were also strongly bound to the active site of tyrosinase. For most analogs, the benzene ring of 2-MBI and the introduced substituents interacted with the amino acids via hydrophobic interactions. The benzene ring interacted with Phe264, Val283, and/or Ala286, whereas the introduced substituents interacted with His61, Phe264, Val283, Ala286, and/or Phe292. In particular, Val283 and Ala286 interacted with most of the analogs via hydrophobic interactions, and the chlorine of **4** formed a hydrogen bond with His61. Interestingly, the benzene ring in **3** was not involved in the hydrophobic interactions; instead, the 5,6-dimethyl substituents interacted with Val283 and Ala286 via hydrophobic interactions. According to the docking simulation results using AutoDock Vina, the binding energies (−5.4~−7.5 kcal/mol) of all analogs were similar to or much more potent than that of kojic acid. However, the 2-mercapto group of analogs was still not involved in the ligand–enzyme interaction. The binding affinities of benzo[*d*]imidazoles lacking the 2-mercapto substituent in the 2-MBI analogs to mushroom tyrosinase were investigated (Figure 5 and Appendix A). Docking simulations of the benzo[*d*]imidazoles were performed using AutoDock Vina. As shown in Appendix A, docking simulation results provided benzo[*d*]imidazoles with binding affinities of −5.9~−7.5 kcal/mol, indicating that the benzo[*d*]imidazoles have slightly greater binding affinities to mushroom tyrosinase than the corresponding 2-MBI analogs **1**–**10**. Four representative benzo[*d*]imidazoles (benzo[*d*]imidazole, 5-nitrobenzo[*d*]imidazole, 5-chlorobenzo[*d*]imidazole, and 5-methylbenzo[*d*]imidazole) were synthesized by reacting 1,2-phenylenediamine derivatives (1,2-phenylenediamine, 4-nitro-1,2-phenylenediamine, 4-chloro-1,2-phenylenediamine, and 4-methyl-1,2-phenylenediamine) with triethyl orthoformate in the presence of sulfamic acid (Figure 5 and Appendix A). Their mushroom tyrosinase activity was evaluated in the presence of l-dopa. Contrary to the docking simulation results, these compounds showed no or very weak tyrosinase inhibitory activity, with IC_50_ values > 400 μM. These results suggest that even though the 2-mercapto substituent of **1**–**10** was not directly involved in the ligand–enzyme interactions, it seemed to contribute to the proper alignment of the remaining structural part (benzo[*d*]imidazole) of the analogs for strong interactions with tyrosinase amino acid residues.

Schrödinger Suite showed that the imidazole ring of 2-MBI analogs plays a principal role in interactions with tyrosinase. In contrast, AutoDock Vina showed that the benzene ring of 2-MBI analogs plays a major role in interactions with tyrosinase. Thus, the mushroom tyrosinase-inhibitory activity of 2-imidazolinethione (Figure 5), which lacks the benzene ring of 2-MBI, was examined to determine whether the benzene ring of 2-MBI plays an important role in tyrosinase inhibition. 2-Imidazolinethione showed only 5% inhibition at 100 μM in the presence of l-dopa (Appendix A), indicating a much lower inhibitory activity than kojic acid. The docking simulation result using AutoDock Vina also showed that 2-imidazolinethione had a very low binding affinity (−3.6 kcal/mol) to mushroom tyrosinase (Appendix A). This result implied that the benzene ring of the 2-MBI analogs played an important role in inhibiting tyrosinase, similar to the results obtained using AutoDock Vina for 2-MBI analogs.

### 3.5. Antioxidant Effects of 2-Mercaptobenzimidazole (2-MBI) Analogs on 2,2-Diphenyl-1-picrylhydrazyl (DPPH) Radical Scavenging, 2,2′-Azino-bis(3-ethylbenzothiazoline-6-sulfonic acid (ABTS) Radical Cation Scavenging, and Reactive Oxygen Species (ROS) Scavenging

As the antioxidant ability of a compound may be involved in the inhibition of melanogenesis [14,15], the antioxidant effects of analogs **1**–**10** on DPPH radicals were explored. Vitamin C (l-ascorbic acid), a strong antioxidant, was used as the positive control. After mixing the test sample and DPPH solutions and standing at room temperature in the dark for 0.5 h, the absorbances of the mixed samples were measured at 517 nm using 1 mM test samples.

Vitamin C showed strong DPPH radical scavenging activity with 98% inhibition, and all the analogs tested exhibited strong scavenging activity against DPPH with more than 65% inhibition (Figure 6A). Six analogs (**2**–**5**, **7**, and **9**) scavenged more than 80% of DPPH radicals, and three analogs (**3**–**5**) showed a similar DPPH radical scavenging activity to vitamin C. The electronic properties of the substituents on the 2-MBI ring did not correlate significantly with the DPPH radical scavenging activity.

The antioxidant potential of **1**–**10** was investigated using the ABTS radical cation scavenging technique. ABTS can be easily oxidized to the corresponding radical cation by oxidants such as potassium persulfate. Because of the correlation between melanogenesis and antioxidant activity, the ABTS radical-scavenging ability of **1**–**10** was examined. A solution of ABTS radical cations was obtained by mixing potassium persulfate with the ABTS solution, and the ABTS radical cation solution was treated with 100 μM test samples. Trolox (Trl) was used as a positive control for activity comparison.

Trl (96% inhibition) exhibited strong ABTS radical cation-scavenging activity, and all analogs showed strong inhibitory activities, with more than 62% inhibition (Figure 6B). Of the 10 analogs, six analogs inhibited the ABTS radical cation by more than 80%, and analog **9**, which showed 90% inhibition, was the strongest ABTS radical scavenger. Considering that all analogs had moderate-to-strong antioxidant capacities, their antioxidant capacity might in part contribute to their anti-melanogenic effects.

Due to the relationship between anti-melanogenesis and antioxidant activity [14,15,32], the ROS-scavenging effects of **1**–**10** were evaluated by measuring the in vitro ROS produced by treatment with 3-morpholinosydnonimine (SIN-1). 2′,7′-Dichlorodihydrofluorescein diacetate (DCFH-DA) is hydrolyzed to 2′,7′-dichlorodihydrofluorescein (DCFH) by esterase, and SIN-1 generates in vitro ROS. Hydrolyzed DCFH reacts with ROS to produce the fluorescent substance dichlorofluorescein (DCF). The ROS-scavenging effects of the analogs were determined by measuring the emitted fluorescence. The in vitro ROS-scavenging effects were measured using 40 μM test samples.

Treatment with SIN-1 significantly enhanced the in vitro ROS levels, and treatment with Trl, a positive control, greatly reduced the in vitro ROS levels enhanced by SIN-1 (Figure 6C). Of the 2-MBI analogs, four (**1** and **4**–**6**) significantly scavenged the in vitro ROS generated. 5-Chloro-2-mercaptobenzo[*d*]imidazole **4** most potently reduced the in vitro ROS levels enhanced by SIN-1, followed by analogs **5**, **6**, and **1**. These results implied that the antimelanogenic effects of these analogs might be partly due to their ROS-scavenging abilities.

### 3.6. Effect of 2-Mercaptobenzimidazole (2-MBI) Analogs on Browning of Freshly-Cut Apple Slices

As the 2-MBI analogs showed high mushroom tyrosinase inhibitory potency, their anti-browning effects on freshly cut apple slices were investigated. Because PTU is poorly soluble in water, the PTU sample was prepared in a DMSO solution. Kojic acid aqueous and PTU DMSO solutions were used as positive controls. Of analogs **1**–**10**, **3** was excluded from this experiment as it is a colored compound that makes measuring the degree of browning difficult. Freshly-cut apple slices were sprayed one time with a 5 mM test sample solution (kojic acid, PTU, or sodium salts of 2-MBI analogs) every 12 h (four-time points: 0, 12, 24, and 36 h) and were stored in an incubator maintained at 20 °C for 48 h. Figure 7 shows the photographs of apple slices with different test sample treatments. According to the photographs of apple slices, at the 12 h storage time point, all test samples inhibited the browning of apple slices compared with the controls. Prolonged storage increased apple slice browning in all groups, but browning in the test sample groups, including the positive controls, was slower than in the controls. Unfortunately, softening (indicated by red arrows) was observed in some apple slices at the 48 h storage time point, and the anti-browning experiment was stopped at the 48 storage time point. The relative browning intensities of the apple slices were determined using CS analyzer software. Because of the softening phenomenon at the 48 h storage time point, photographs of apple slices stored for 36 h were used to measure the relative browning intensities. The results are shown in Figure 8A. Kojic acid inhibited browning by 7% compared to the control, and all 2-MBI analogs exhibited more potent inhibition (19.3–36.1% inhibition) of apple slice browning than kojic acid. In addition, six analogs (**2** and **4**−**8**) showed similar anti-browning activity to PTU (21.9% inhibition), and three analogs (**1**, **9**, and **10**) with 28.5–36.1% inhibitions inhibited apple slice browning more potently than PTU.

CIE color values were measured using a CR-10 spectrophotometer [33]. CIE color values have been widely used to evaluate the degree of browning in fruits and vegetables [34]. With increasing storage time, the b* (yellowness and blueness) and L* (lightness) values decreased, whereas the a* (redness and greenness) values increased. The overall color difference (Δ*E*) of apple slices showed an increasing pattern with increased storage time (Figure 8B). PTU exhibited a stronger anti-browning potential than kojic acid regarding the outcome of Δ*E*. However, because of its low water solubility, PTU is difficult to use as an anti-browning agent for fruits and vegetables. Analog **10** had the lowest increase in Δ*E*, indicating that **10** has the most effective potential for preventing the browning of apple slices. According to the results of Δ*E* values, all analogs except **7** showed similar or better anti-browning potential than kojic acid and PTU at the 12 and 24 h storage time points. In addition, at the 36 h storage time point, all analogs except **7** exhibited more potent anti-browning potential than kojic acid, and six analogs (**2**, **4**, **6**, and **8**–**10**) effectively inhibited the browning of apple slices more strongly than PTU. Due to softening of some apple slices at 48 h storage time, Δ*E* values of **2**, **4**, **5**, **8**, and **9** steeply increased at 48 h storage time. Specifically, Δ*E* values of **5** and **9** at 48 h storage time increased the most due to the softening phenomenon in a large region of apple slices. Analog **10** showed the lowest Δ*E* value, indicating that it inhibited the browning of apple slices the most. It is believed that the softness of apples is due to either the apple itself or the storage environment. Therefore, in addition to **1** and **10**, analogs **2**, **4**, **5**, **8**, and **9** can be considered to have sufficient anti-browning potential on apple slices if the softening factor of the apple slices is neglected. Figure 8C presents the changes of Δ*E* values for representative analogs **1**, **9**, and **10** showing strong anti-browning efficacy in apple slices. In contrast, L* values in all groups decreased with increasing storage time, and the control group showed a large L* value change (ΔL* = 7.5) at the 48 h storage time point (Figure 8D). Among the 2-MBI analogs, **5** showed the largest L* value change (ΔL* = 8.2) at the 48 h storage time point due to the softening of apple slices over a wide area. PTU showed a smaller change in L*value (ΔL* = 3.4) than kojic acid (ΔL* = 6.4). The L* value change of **10** (ΔL* = 3.1) was similar to that of PTU, and analog **7** showed the smallest L* value change (ΔL* = 1.3). Figure 8E presents the changes in L* values for **1**, **9**, and **10**, which showed strong anti-browning efficacy in apple slices. These results suggest that 2-MBI analogs, including **1**, **2**, **4**, and **8**–**10**, have the potential to delay the browning of freshly cut apple slices effectively.

### 3.7. Cytotoxicity of 2-Mercaptobenzimidazole (2-MBI) Analogs in Human Embryonic Kidney Cells (HEK-293) Cells

Safety is an important issue in the application of 2-MBI analogs as an anti-browning agent. The cytotoxicity of 2-MBI analogs was evaluated at four different concentrations (0, 2.5, 10, and 40 μM). For this purpose, an EZ-Cytox assay was performed using HEK-293 cells. As shown in Figure 9, although analogs **3** and **6** were weakly and moderately cytotoxic at 40 μM, respectively, the remaining analogs showed no perceptible cytotoxicity until the maximum concentration was tested. In the case of analog **6**, the IC_50_ values against mushroom tyrosinase were 0.01 and 0.02 μM in monophenolase and diphenolase, respectively, so its toxic concentration was 1000–2000 times higher than the IC_50_ value of tyrosinase inhibition. In other words, the SI (selectivity index) value of **6** is very high, so there is no problem with potential health risk and practicability. These results suggest that 2-MBI analogs, such as **1**, **2**, **4**, **6** and **8**–**10**, may be safe and potent anti-browning agents for vegetables and fruits.

## 4. Conclusions

In order to discover safe and effective anti-browning agents, 2-MBI analogs were synthesized because the sulfhydryl substituent plays a key role in tyrosinase inhibition and acts as an antioxidant. Most 2-MBI analogs showed very low IC_50_ values (<0.8 μM) against mushroom tyrosinase for both monophenolase and diphenolase activities. Kinetic studies using mushroom tyrosinase indicated that **6** and **10** were competitive inhibitors, whereas **4** was a mixed-type inhibitor, supported by the docking simulation. The 2-mercapto substituent in 2-MBI analogs plays an important role in tyrosinase inhibition. Most of the 2-MBI analogs effectively retarded the browning of freshly cut apple slices during storage.

## Data Availability

Data will be made available on request.

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
