# Peer review of "Anti-Browning Effect of 2-Mercaptobenzo[d]imidazole Analogs with Antioxidant Activity on Freshly-Cut Apple Slices and Their Highly Potent Tyrosinase Inhibitory Activity"

_antioxidants, 2023, doi:10.3390/antiox12101814_

Round 1

Reviewer 1 Report

The work is interesting. The organic synthesis seems correct and the

 work deserves publication if the authors can answer my question.

The activity tests last 30 minutes and the authors measure the   absorbance of dopachrome at 475 nm. In my opinion, they have not   taken into account two aspects: a) that dopachrome is unstable and b)   that thiols can react with o-quinones.

Author Response

The work is interesting. The organic synthesis seems correct and the work deserves publication if the authors can answer my question.

The activity tests last 30 minutes and the authors measure the   absorbance of dopachrome at 475 nm. In my opinion, they have not   taken into account two aspects: a) that dopachrome is unstable and b)   that thiols can react with o-quinones.

Answer:

Thank you for your valuable comments.

Regarding a) dopachrome is unstable:

Although we measured the absorbance of dopachrome for 30 minutes, the data we used for the inhibitory activity test was the portion with the largest change in absorbance per unit time. In other words, when creating a graph for time (x-axis) and absorbance (y-axis), the part with the largest slope was used as the inhibitory active data. In other words, rather than using the change in dopachrome over 30 minutes, the data used was the instantaneous change in dopachrome at a specific point in time within 30 minutes. Therefore, the instability of dopachrome is not a problem in measuring the instantaneous change in dopachrome.

Regarding b) thiols can react with o-quinones.

Because cysteine contains primary thiol, it reacts quickly with o-quinones and leads to the pathway to synthesize pheomelanin. 2-MBI compound also has a thiol structure, but because the thiol of 2-MBI is secondary, it causes greater steric hindrance than primary thiol. Therefore, the thiol of 2-MBI reacts relatively poorly with o-quinones. In addition, 2-MBI thiol can form a tautomer structure with an imidazole ring, so the thiol quickly equilibrates with the thione structure, so its nucleophilicity is not as high as that of common thiols such as cysteine.

Reviewer 2 Report

The manuscript entitled 'Anti-browning effect of 2-mercaptobenzo[d]imidazole analogs with antioxidant activity on freshly-cut apple slices and their highly potent tyrosinase inhibitory activity' explores a new approach of improving the color retention of apple slices by using 2-MBI as a tyrosinase inhibitor. The experiments were well-designed, and the presented results are in good quality. However, there are still some concerns and questions that the authors should consider before the paper is considered acceptable:

1. In the introduction, it is recommended that the authors add more justification of using 2-MBI compounds and analogs as potential antioxidant agent for fruits. 

2. Section 2.11: in the methodology section, please add the necessary description of mean comparison tests when choosing p value lower than 0.01 and 0.001 as statistically significant. As they show up in the results.

3. The cytotoxic testing draws some some concerns. 2-MBI is generally advised as use against foods or pharmaceuticals. The results show that several analogs in this study show cytotoxic effect, and are concentration related. Would that cause potential health risks if abused? And would that affect its practicability?

Ok

Author Response

Reviewer 2

The manuscript entitled 'Anti-browning effect of 2-mercaptobenzo[d]imidazole analogs with antioxidant activity on freshly-cut apple slices and their highly potent tyrosinase inhibitory activity' explores a new approach of improving the color retention of apple slices by using 2-MBI as a tyrosinase inhibitor. The experiments were well-designed, and the presented results are in good quality. However, there are still some concerns and questions that the authors should consider before the paper is considered acceptable:

  1. In the introduction, it is recommended that the authors add more justification of using 2-MBI compounds and analogs as potential antioxidant agent for fruits. 

Thank you for your valuable suggestion.

Following the reviewer's suggestion, the following sentences were added to the Introduction: “Typically, fruits undergo a slow process of oxidation after harvest, which produces undesirable flavors and often reduces the quality of the fruit. Due to the antioxidant ability of the mercapto functional group, applying 2-MBI analogs to fruits is likely to not only suppress browning by inhibiting tyrosinase activity, but also help improve fruit quality through antioxidant effects.”

  1. Section 2.11: in the methodology section, please add the necessary description of mean comparison tests when choosing p value lower than 0.01 and 0.001 as statistically significant. As they show up in the results.

Thank you for your valuable comments.

We used the same mean comparison tests in all cases (p < 0.05, p <0.01, and p <0.001). In ‘Two-sided p-value < 0.05 was considered statistically significant’, ‘two-sided p-value < 0.05’ means all instances of p < 0.05, p <0.01, and p <0.001.

  1. The cytotoxic testing draws some some concerns. 2-MBI is generally advised as use against foods or pharmaceuticals. The results show that several analogs in this study show cytotoxic effect, and are concentration related. Would that cause potential health risks if abused? And would that affect its practicability?

Thank you for your insightful and valuable comments

Of the ten 2-MBI analogs, only two analogs (3 and 6) showed mild cytotoxicity at 40 μM, the highest concentration tested. In the case of analog 6, the IC50 value is 0.01 and 0.02 μM in monophenolase and diphenolase, respectively, so the toxic concentration is 2000-1000 times higher than the IC50 value. In other words, the SI (selectivity index) value is very high, so there is no problem with potential health risk and practicability. It is thought that however, in the case of analog 3, the SI value is relatively low (SI = 53), so there is a possibility that it may cause problems with potential health risk and practicability. However, there are eight analogs that did not show toxicity up to the highest concentration (40 μM) tested. Therefore, most MBI analogs are thought to be applicable to foods or pharmaceuticals without any problems.